# An Evaluation of Neuron-Specific Enolase as a Biomarker of Neurological Impact in Pacemaker-Implanted Patients with Atrial High-Rate Episodes: An Observational Study from Turkey

**DOI:** 10.3390/medicina61020324

**Published:** 2025-02-12

**Authors:** Ahmet Cinar, Omer Gedikli, Muhammet Uyanik, Bahattin Avci, Ozlem Terzi

**Affiliations:** 1Department of Cardiology, Faculty of Medicine, Ondokuz Mayıs University, Samsun 55270, Turkey; drgedikli@hotmail.com; 2Clinical of Cardiology, Carsamba State Hospital, Samsun 55500, Turkey; muhammetuyanik@hotmail.com; 3Department of Medical Biochemistry, Faculty of Medicine, Ondokuz Mayıs University, Samsun 55270, Turkey; bahattinavci@hotmail.com; 4Department of Public Health, Faculty of Medicine, Ondokuz Mayıs University, Samsun 55270, Turkey; ozlem.terzi@omu.edu.tr

**Keywords:** atrial high-rate episodes (AHRE), silent neurological ischemia, neuron-specific enolase, pacemaker

## Abstract

*Background and Objectives:* An atrial high-rate episode (AHRE) is defined according to the European Society of Cardiology (ESC) guidelines as a heart rate of ≥175 bpm lasting at least 5 min. This study aimed to evaluate whether neuron-specific enolase (NSE) levels, an indicator of neurological impact, could serve as a surrogate biomarker for silent neurological ischemia in patients with atrial high-rate episodes (AHREs). *Materials and Methods:* Patients with AHRE detected in a pacemaker analysis and a control group without any arrhythmias were included. Patients with AHRE were divided into subgroups according to AHRE duration—Group 1: AHRE < 5 min, Group 2: AHRE ≥ 5 min–<1 h, Group 3: AHRE ≥ 1 h–<24 h, Group 4: AHRE ≥ 24 h. Neuron-specific enolase (NSE) levels were measured using a double-antibody enzyme-linked immunosorbent assay (ELISA) with a sensitivity of 0.05 ng/mL. Imaging techniques were not employed in this study, and NSE was used as an indirect measure of potential neurological impact. *Results:* There were 160 patients, including 80 (50.0%) in the AHRE group and 80 (50.0%) in the control group. According to AHRE duration, there were 24 (30.0%) patients in Group 1, 33 (41.2%) in Group 2, 19 (23.8%) in Group 3, and 4 (5.0%) in Group 4. Patients with AHRE had statistically significant differences in age, sPAP, transmitral E/A ratio, and NSE levels. The mean NSE levels of all groups were significantly different (*p* < 0.001). A correlation analysis in patients with AHRE showed a very strong positive correlation between AHRE duration and NSE values as well as correlations with age, virtual CHA_2_DS_2_-VASc score, and LA diameter. NSE levels were positively correlated with AHRE duration and LA diameter. AHRE duration was an independent predictor of elevated NSE levels. *Conclusions:* It was shown that AHRE is associated with silent neurological ischemia and that NSE levels can be used to demonstrate these neurological effects. Future studies can contribute to the development of more effective treatment strategies based on these findings by investigating the neurological effects of AHRE in more detail.

## 1. Introduction

Atrial fibrillation (AF) is a commonly encountered arrhythmia among cardiovascular diseases, with its prevalence increasing with age [1]. AF is a significant contributor to cardiovascular morbidity and mortality, making it a prominent topic of research in the field of health. Furthermore, AF is known to have a strong association with ischemic stroke. Traditionally, AF is diagnosed through electrocardiography (ECG); however, with advancements in technology, this condition can now be detected earlier through implantable cardiac devices and wearable technologies [2].

In recent years, atrial high-rate episodes (AHREs) have been defined as tachyarrhythmic events detectable only by implantable cardiac devices. AHRE is often asymptomatic, and there is variability in definitions and durations across studies. According to the 2024 European Society of Cardiology (ESC) guidelines, AHRE is defined as a heart rate of ≥175 bpm lasting at least 5 min [3]. This definition links AHRE to cardiovascular events such as AF and ischemic stroke, though further research is needed to confirm this association.

Recent studies have demonstrated that AHRE is associated with silent neurological ischemia and a decline in cognitive functions [4,5]. Silent stroke, often asymptomatic, is a significant condition that can lead to cognitive impairment over time. Although AHRE may not cause silent neurological ischemia to the same extent as AF, it has been found to be related to cognitive decline and other neurological disorders [6].

This study aims to investigate whether neuron-specific enolase (NSE) levels, an indicator of silent neurological ischemia, can serve as an effective biomarker to demonstrate silent neurological ischemia without the use of imaging techniques and to examine factors associated with AHRE. While NSE is a biochemical marker associated with neuronal damage, it does not provide direct evidence of silent neurological ischemia without accompanying imaging studies. This study seeks to explore NSE as a potential surrogate marker for such ischemia, recognizing the need for future imaging-based studies to validate these findings.

## 2. Methods

This study was designed as a single-center prospective study. A total of 160 patients were included in the study, consisting of 80 patients with previously implanted permanent pacemakers who were identified with atrial high-rate episodes (AHREs) during pacemaker interrogation and 80 patients with no arrhythmias detected during pacemaker interrogation. These patients were recruited from the Cardiology Arrhythmia Outpatient Clinic at Ondokuz Mayıs University’s Faculty of Medicine between January 2021 and June 2022. Patients with detected AHRE were further classified into subgroups based on the duration of AHRE—Group 1: AHRE < 5 min, Group 2: AHRE ≥ 5 min–<1 h, Group 3: AHRE ≥ 1 h–<24 h, Group 4: AHRE ≥ 24 h.

All participants were provided with detailed information about the study, and their written and verbal informed consent were obtained. The study was ethically approved by the Ondokuz Mayıs University’s Clinical Research Ethics Committee on 16 June 2020 (Approval No. OMU KAEK 2020/390). Financial support was provided by Ondokuz Mayıs University under the project code PYO.TIP.1904.20.013.

### 2.1. Inclusion Criteria for the Study 

Male and female patients aged 18 years or older who had permanent pacemakers implanted within the last six months were included.

### 2.2. Exclusion Criteria for the Study

Exclusions included patients under 18 years of age; patients who did not provide consent; history of neurological malignancies or tumors with neurological metastases; history of neurodegenerative diseases (e.g., Multiple Sclerosis, Alzheimer’s, Parkinson’s disease); known history of atrial fibrillation (AF); use of oral anticoagulants; the presence of an active infection; history of intracranial events; history of head trauma; the detection of arrhythmias other than AHRE during pacemaker interrogation; the presence of a thrombus or significant spontaneous echo contrast (SEC) in cardiac chambers on echocardiography; a left ventricular ejection fraction below 50% or left atrial diameter exceeding 40 mm on echocardiography; and pacemakers with implantable cardioverter-defibrillator (ICD) or cardiac resynchronization therapy (CRT) features.

Electrocardiographic evaluations for all participants were conducted using a Biolight E70 ECG device (Guangdong Biolight Meditech Co., Ltd., Zhuhai, Guangdong Province, China) before pacemaker interrogation. Only patients with permanent pacemakers implanted within the last six months were included in the study. Devices with implantable cardioverter-defibrillator (ICD) or cardiac resynchronization therapy (CRT) capabilities were excluded. Patients with permanent pacemakers were assessed using the Medtronic Corelink Encore™ 29901 (Minneapolis, MN, USA) pacemaker programming device. During pacemaker interrogation, all patients were evaluated for proper device function, including battery status, lead impedance, sensing thresholds, and pacing thresholds. Patients with lead dislodgement, sensing defects, or battery issues were excluded from the study. False AHREs were differentiated from genuine episodes by a careful analysis of device logs and patient history. The neurological histories of the patients were reviewed. After applying inclusion and exclusion criteria, patients with AHRE within the last 72 h were evaluated. Patients with other conditions causing elevated NSE levels were excluded. Genuine-AHRE patients underwent echocardiographic evaluations using a Vivid E9 device (GE Vingmed Ultrasound, Horten, Norway).

Similar procedures were applied to the control group, consisting of permanent pacemaker patients meeting inclusion and exclusion criteria with no arrhythmias detected.

Venous blood samples were collected from the brachial vein of the forearm in all patients. Complete blood count, renal function tests, and NSE levels were analyzed. All samples were processed at the Ondokuz Mayıs University’s Faculty of Medicine’s Biochemistry Department Laboratory. NSE levels were measured using commercial test kits provided by Bioassay Technology Laboratory Company through a double-antibody enzyme-linked immunosorbent assay. Sensitivity was 0.05 ng/mL, with a measurement range of 0.1 ng/mL to 40 ng/mL.

### 2.3. Statistical Analysis

Data collected from the study were coded and analyzed using SPSS (Statistical Package for the Social Sciences, Version 22 for Windows, SPSS Inc., Chicago, IL, USA). Continuous variables were expressed as the mean ± standard deviation for parametric data or as the median (minimum–maximum) for non-parametric data. Frequency data were expressed as numbers and percentages. The normality of distribution for all measurable variables was assessed using the Kolmogorov–Smirnov test. In the evaluation, it was determined that all variables, except for hemoglobin and platelet count, did not follow a normal distribution. Comparisons of frequency data were conducted using Pearson’s Chi-Square test and Fisher’s Exact test. For continuous variables, comparisons between groups were performed using Student’s *t*-test and ANOVA for parametric data and the Mann–Whitney U test and Kruskal–Wallis test for non-parametric data. For variables with significant differences among two or more groups, Bonferroni-corrected Mann–Whitney U tests were performed to identify the contributing group. Correlations between groups were assessed using Spearman correlation tests due to the non-normal distribution of certain variables. The strength of the correlation was interpreted as follows: r = 0.00–0.24 (weak), r = 0.25–0.49 (moderate), r = 0.50–0.74 (strong), and r = 0.75–1.00 (very strong). Single and multivariate linear regression analyses were conducted to identify factors influencing NSE levels. Single and multivariate logistic regression analyses were performed for factors associated with AHRE duration. Statistical significance was considered at *p* < 0.05 for all tests.

The sample size was calculated using PASS 11 software (Power and Sample Size, version 11). The calculation was based on data from previous studies that examined differences in NSE levels between patients with arrhythmias and control groups. According to these studies, a total of 78 participants was required to achieve 80% power (1 − β = 0.80) at a 5% significance level (α = 0.05). To account for potential dropouts or exclusions, 80 patients per group were included, resulting in a total of 160 participants.

Neuron-specific enolase (NSE) levels were evaluated based on previously reported baseline and pathological threshold values in the literature. Specifically, studies examining the use of NSE as a biomarker for ischemic events were referenced [7,8].

## 3. Results

A total of 133 patients who had a permanent pacemaker implanted within the last 6 months and met the inclusion and exclusion criteria were included in the study. Of these, 45 patients were excluded due to no AHRE detected within the last 72 h, and 8 patients were excluded due to atrial sensing defects causing false AHRE. Ultimately, 80 patients with detected AHRE were included. For the control group, 91 patients with a permanent pacemaker implanted within the last 6 months and meeting the inclusion and exclusion criteria, without any arrhythmia detected in the pacemaker analysis, were initially considered. However, 11 patients were excluded due to unclassifiable palpitations, leaving a total of 80 patients in the control group. In total, 160 volunteer patients with permanent pacemakers were included in the study.

The patient group consisted of 36 women (45.0%) and 44 men (55.0%), while the control group included 32 women (40.0%) and 48 men (60.0%) (*p* = 0.522). The mean age of the patient group was 67.49 ± 16.6 years, compared to 65.00 ± 13.1 years in the control group (*p* = 0.046). The baseline demographic and clinical characteristics of the patients included in the study are presented in Table 1.

When the echocardiographic parameters of the patients were evaluated, statistically significant differences were found between the two groups in terms of sPAB and the transmitral E/A ratio. The mean sPAB value in the patient group was 27.16 ± 10.51, while in the control group, it was 23.74 ± 6.86 (*p* = 0.049). The transmitral E/A ratio was 1.38 ± 0.23 in the patient group and 1.12 ± 0.35 in the control group, which was statistically significant (*p* < 0.001) (Table 2).

Patients with AHRE were then divided into subgroups. Group 1 (<5 min) included 24 patients (30.0%), Group 2 (≥5 min–<1 h) included 33 patients (41.2%), Group 3 (≥1 h–<24 h) included 19 patients (23.8%), and Group 4 (≥24 h) included 4 patients (5.0%) (Figure 1).

When the groups were examined according to NSE levels, there was a statistically significant difference in the median NSE values (*p* ≤ 0.001). A further analysis (Bonferroni-corrected Mann–Whitney U test) showed that when all groups were compared with each other, a statistically significant difference was found between AHRE duration and the median NSE values (*p* ≤ 0.001) (Figure 2).

In the correlation analysis conducted in patients with AHRE, a statistically significant and very strong positive correlation was found between AHRE duration and NSE values (r = 0.842; *p* ≤ 0.001). A statistically significant correlation was also observed between AHRE duration and age (r = 0.235; *p* = 0.036), virtual CHA_2_DS_2_-VASc score (r = 0.226; *p* = 0.044), and LA diameter (r = 0.300; *p* = 0.007) (Table 3) (please see the Appendix A for the graph).

In patients with AHRE, the correlation analysis of NSE levels revealed a statistically significant, strongly positive relationship with AHRE duration (r = 0.842; *p* < 0.001). Additionally, a weak but statistically significant positive correlation was found between NSE levels and LA diameter (r = 0.242; *p* = 0.030) (Table 4) (please see the Appendix A for the graph). Univariate and multivariate linear regression analyses were performed to identify factors affecting NSE levels. AHRE duration was an independent predictor of NSE elevation, with an odds ratio (OR) of 5.28 (95% confidence interval [CI]: 4.02–6.54; *p* < 0.001) (Table 5).

## 4. Discussion

Atrial high-rate episodes (AHREs) are arrhythmic events that can be detected by cardiac implantable devices and are gaining increasing clinical significance. Our study investigated the relationship between AHRE and neuron-specific enolase (NSE) levels, a biochemical marker of neurological complications, and our results demonstrate the biochemical manifestation of AHRE’s neurological effects. The relationship between silent ischemia and NSE has been studied and demonstrated in the previous literature [9,10]. In this regard, our study examined the potential of NSE levels as a biomarker for silent neurological ischemia in patients with AHRE. A strong positive correlation between NSE levels and AHRE duration suggests that AHRE may be linked to cerebral ischemic processes. However, as NSE is not a definitive marker of silent neurological ischemia, our findings should be interpreted with caution. Instead, NSE levels may serve as a potential surrogate marker for such ischemia in patients with AHRE.

Previous studies on the neurological effects of AHRE have emphasized its strong association with ischemic stroke [11,12,13]. However, the relationship between AHRE and silent neurological ischemia has gained more attention in recent years. Specifically, the often asymptomatic nature of AHRE may cause patients’ neurological effects to progress unnoticed. This could lead to the underappreciation of AHRE’s long-term cardiovascular effects. In our study, a significant increase in NSE levels was observed with increasing AHRE duration, indicating that prolonged AHRE may enhance the severity of its neurological effects. This finding underscores the importance of an early diagnosis and treatment of AHRE.

Our results revealed that age showed a significant correlation with both AHRE duration and NSE levels. This could be explained by the increase in atrial remodeling, changes in cardiac substrates, and the decline in the neurological reserve with age [14,15]. Particularly in older individuals, longer AHRE duration poses a greater risk for cerebrovascular events. It has been demonstrated that AHRE duration increases NSE levels, thus preparing the base for neurological complications in elderly patients [16,17]. Our study confirms these findings and highlights the need for further research into age-related effects of AHRE.

Age directly affects many parameters of cardiovascular disease. In our study, although no significant correlation was found between age and virtual CHA_2_DS_2_-VASc scores or AHRE duration, atrial remodeling and diastolic dysfunction mechanisms in elderly individuals could explain the prolonged AHRE duration [17]. The literature suggests that AHRE is more frequent and longer in older individuals [18,19].

Echocardiographic findings also highlight the importance of considering the hemodynamic effects of AHRE. Specifically, the relationship between left atrial (LA) diameter and AHRE duration (r = 0.300, *p* = 0.007) points to the role of atrial remodeling in the development of AHRE. An increase in LA diameter may create a substrate that supports AHRE and, consequently, increase the risk of cerebrovascular events [20]. The expansion of LA diameter contributes to atrial remodeling and the formation of arrhythmogenic substrates, facilitating AHRE development.

No temporal study has been found in the literature regarding the time after an AHRE and the increased risk of stroke. Given that the half-life of NSE levels is 48–72 h, the timing of silent neurological ischemia post-AHRE could be significantly informed by our study, as it focuses on events occurring within 72 h of the last AHRE.

Patients with ICDs and CRTs typically have lower EF heart failure, larger LA sizes, and a history of AF or other arrhythmias [21,22]. Therefore, AHRE alone may not sufficiently demonstrate its role in ischemia, and our study only included patients with pacemakers to focus on this relationship. This approach helps clarify the correlation between asymptomatic AHRE and ischemic events more distinctly.

High diastolic blood pressure is associated with reduced RV (right ventricular) function [23,24]. Systolic pulmonary artery pressure (sPAP) and transmitral E/A ratios were significantly higher in the AHRE group (*p* = 0.049). The relationship between AHRE duration and pulmonary artery pressure can be explained by impaired atrioventricular synchronization, which indirectly increases pulmonary vascular resistance. This finding suggests that AHRE may not only contribute to atrial but also to ventricular load.

In our multivariate analysis, the relationship between AHRE duration and NSE levels was independently confirmed (OR = 5.28; *p* < 0.001). This finding suggests that the neurological effects of AHRE are independent of other cardiovascular risk factors. This highlights that AHRE is not only an atrial rhythm disturbance but also a risk factor for increasing neurological ischemia. The literature has also shown that AHRE, especially when prolonged, can increase the risk of stroke [25,26]. Our study suggests that the duration of AHRE may be a prognostic parameter, with longer durations associated with a higher risk of neurological damage, as indicated by the significant increase in NSE levels (OR = 5.28; *p* < 0.001). The parallel increase in NSE levels with AHRE duration may require more frequent neurological monitoring of patients with AHRE.

Our study emphasizes the importance of risk stratification in the clinical management of AHRE. In patients categorized by AHRE duration, those with episodes lasting more than 24 h were found to require neurological evaluation. The use of biochemical markers like NSE for the early detection of silent cerebrovascular events may be beneficial for personalizing patient management. Moreover, few studies specifically examine the relationship between AHRE duration thresholds and neurological complications, and our study makes a significant contribution to the literature in this regard. While the timing and initiation of anticoagulation therapy to mitigate stroke risk remain unclear, recent studies such as the NOAH-AFNET 6 and ARTESIA trials aim to address this issue [27,28]. In the NOAH-AFNET study, no significant difference in cardiovascular death, stroke, and systemic embolism rates was observed between the edoxaban and placebo groups, although bleeding rates were higher in the edoxaban group. The ARTESIA study demonstrated that apixaban was associated with a lower risk of stroke or a systemic embolism than aspirin in patients with subclinical atrial fibrillation, though major bleeding risks were higher [29]. More definitive studies on anticoagulation therapy are needed in the future.

### Study Limitations

This study has several limitations. First, the patient population was drawn from a single center, which limits the generalizability of the results. Since NSE does not provide information about silent neurological ischemia older than 2–3 days, we were only able to include patients with AHRE detected within the last 72 h, thus identifying those who might develop new silent neurological ischemia. In this study, NSE levels were measured only within 72 h following the last detected AHRE to evaluate acute neurological effects. However, we did not measure NSE levels during AHRE-free periods, which could provide valuable baseline information regarding neurological ischemic activity. Future studies incorporating serial measurements during and outside AHREs could better clarify the temporal association between AHRE and neurological ischemia. A significant limitation of this study is the absence of neuroimaging to confirm silent neurological ischemia. NSE levels, while indicative of neuronal injury, cannot establish a direct causal relationship with brain injury. Therefore, our findings should be interpreted with caution, and future studies incorporating imaging modalities, such as MRI, are essential to corroborate these results. Medtronic-brand pacemaker devices were used in this study. Different devices may have varying parameters and algorithms, which could limit the generalizability of the results. Additionally, only patients with permanent pacemakers were included; patients with ICD and CRT devices were not part of this study. Future studies should investigate AHRE in broader patient populations and with different devices. The number of patients with AHRE lasting ≥24 h was limited to four. Cognitive events and imaging methods were not employed during follow-up, and clinically significant strokes could not be evaluated. Future follow-up studies may assess the relationship between AHRE, clinically significant strokes, and cognitive functions.

## 5. Conclusions

This study demonstrates an association between AHRE and increased NSE levels, suggesting potential neurological impact. However, NSE should be considered a surrogate marker rather than definitive evidence of silent neurological ischemia. Further research incorporating neuroimaging is necessary to confirm these associations and clarify the role of NSE in this context.

## Figures and Tables

**Figure 1 medicina-61-00324-f001:**
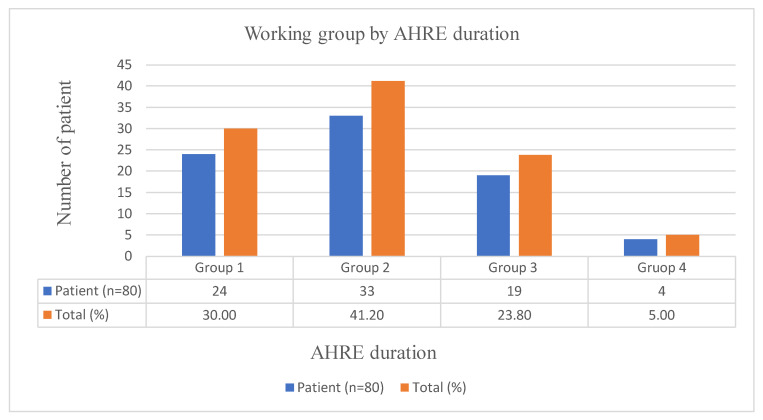
Working groups by AHRE duration.

**Figure 2 medicina-61-00324-f002:**
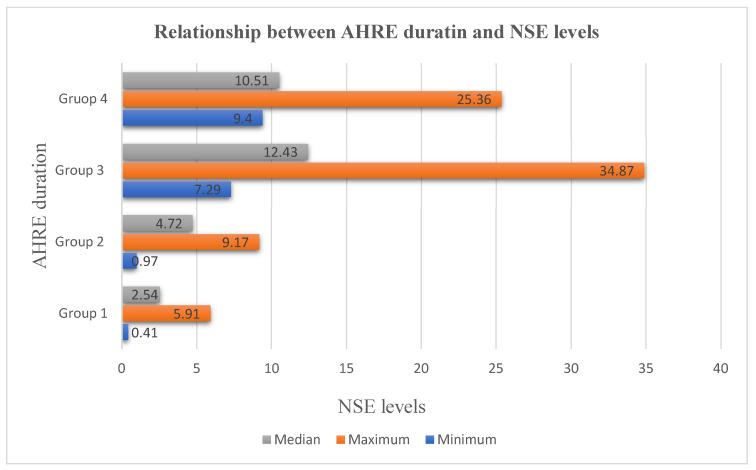
Relationship between AHRE duration and NSE levels.

**Table 1 medicina-61-00324-t001:** Baseline Demographic and Clinical Characteristics of the Patients.

Characteristics	Patients (*n* = 80)	Control (*n* = 80)	*p* Values
Age—years	67.49 (±16.6)	65.00 (±13.1)	0.046
Female sex—no. (%)	36 (45.0)	32 (40.0)	0.522
Mean virtual CHA_2_DS_2_-VASc score—IQR	2.5 (0–5)	2.5 (0–5)	0.950
History of hypertension—no. (%)	54 (67.5)	59 (73.8)	0.385
History of PCI—no. (%)	5 (6.3)	6 (7.5)	0.755
History of CABG—no. (%)	3 (3.8)	5 (6.3)	0.468
Peripheral arterial disease—no. (%)	3 (3.8)	4 (5.0)	0.699
Diabetes mellitus—no. (%)	18 (22.5)	23 (28.8)	0.365
Hyperlipidemia—no. (%)	13 (16.3)	16 (20.0)	0.538
Chronic renal failure—no. (%)	8 (10.0)	3 (3.8)	0.118
Hemoglobin—g/dL	13.51 (±1.60)	13.77 (±1.46)	0.289
Platelet—thousand/μL	233.65 (±56.45)	218.10 (±60.55)	0.095
Estimated glomerular filtration rate—mL/min	86.01 (±20.4)	80.63 (±21.6)	0.472
NSE (neuron-specific enolase)—ng/mL	5.02 (±6.56)	3.02 (±1.60)	<0.001
Medical treatment—no. (%)			
Antiplatelet	33 (41.3)	38 (47.5)	0.426
Beta-blocker	34 (42.5)	37 (46.4)	0.064
Angiotensinogen-converting enzyme inhibitor	13 (16.4)	13 (16.4)	0.475
Angiotensin receptor blocker	23 (27.6)	21 (20.1)	0.176
Mineralocorticoid receptor antagonist	7 (8.8)	14 (17.5)	0.101
Calcium channel blocker	7 (8.8)	8 (10.0)	0.786
Statin	15 (18.8)	20 (25.0)	0.339

CABG: Coronary artery bypass graft; PCI: Percutaneous coronary intervention.

**Table 2 medicina-61-00324-t002:** Comparison of echocardiography parameters between groups.

Parameters	Patients (*n* = 80)	Control (*n* = 80)	*p* Values
LVEF (%)	57.13 (±4.65)	56.53 (±5.16)	0.390
LVEDD (mm)	46.81 (±4.87)	45.63 (±3.38)	0.075
LVESD (mm)	33.35 (±4.95)	32.41 (±3.83)	0.300
LA (mm)	35.60 (±2.54)	35.40 (±2.56)	0.478
IVS (mm)	11.29 (±2.08)	11.24 (±1.77)	0.982
PW (mm)	10.90 (±1.76)	10.54 (±1.38)	0.153
E/A	1.38 (±0.23)	1.12 (±0.35)	<0.001
sPAP (mmHg)	27.16 (±10.51)	23.74 (±6.86)	0.049

E/A: Early Diastolic Flow Rate/Late Diastolic Flow Rate; IVS: Interventricular Septum; LA: Left Atrium; LVEDD: Left Ventricle End-Diastolic Diameter; LVEF: Left Ventricular Ejection Fraction; LVESD: Left Ventricle End-Systolic Diameter; PW: Posterior Wall; sPAP: Systolic Pulmonary Artery Pressure.

**Table 3 medicina-61-00324-t003:** Spearman correlation analysis results related to AHRE duration.

	r	*p* Value
NSE levels	0.842	<0.001 ***
Age	0.235	0.036 *
Virtual CHA_2_DS_2_-VASc	0.226	0.044 *
LVEF	−0.163	0.149
LA	0.300	0.007 **
sPAP	0.119	0.292

*: *p* < 0.05—significant; **: *p* < 0.01—highly significant; ***: *p* < 0.001—very highly significant. LVEF: Left Ventricular Ejection Fraction; LA: Left Atrium; sPAP: Systolic Pulmonary Artery Pressure.

**Table 4 medicina-61-00324-t004:** Spearman correlation analysis results related to NSE levels.

	r	*p* Value
AHRE duration	0.842	<0.001 ***
Age	0.166	0.142
Virtual CHA_2_DS_2_-VASc	0.156	0.168
LVEF	−0.145	0.198
LA	0.242	0.030 *
sPAP	0.077	0.494

*: *p* < 0.05—significant; ***: *p* < 0.001—very highly significant. AHRE: Atrial High-Rate Episode; LVEF: Left Ventricular Ejection Fraction; LA: Left Atrium; sPAP: Systolic Pulmonary Artery Pressure.

**Table 5 medicina-61-00324-t005:** Univariate and multivariate linear regression analysis on predictors of NSE level.

	Univariate Analysis		Multivariate Analysis	
	OR (%95 CI)	*p* Value	OR (%95 CI)	*p* Value
AHRE duration	5.41 (4.21–6.62)	<0.001	5.28 (4.02–6.54)	<0.001
LA	0.34 (0.27–0.66)	0.168	0.16 (−0.26–0.58)	0.456

AHRE: Atrial High-Rate Episode; LA: Left Atrium; OR: Odds Ratio.

## Data Availability

The data presented in this study are available on request from the corresponding author.

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
