# Peer review of "An Evaluation of Neuron-Specific Enolase as a Biomarker of Neurological Impact in Pacemaker-Implanted Patients with Atrial High-Rate Episodes: An Observational Study from Turkey"

_medicina, 2025, doi:10.3390/medicina61020324_

Round 1
Reviewer 1 Report
Comments and Suggestions for Authors
Cinar et al. performed a study to assess the silent neurological infarcts in patients with pacemaker with AHRE.
My recommendations are:
There is a significant limitation in the study. The authors report that they will investigate silent neurological ischemia, but they never performed imaging, only NSE, which cannot be considered as a corresponding direct association with certainty of brain injury. In this way, there are some inconsistencies with the title, abstract, object, and methodology.
The title should contain the type of article being published; and include the country or region where it was done. The title is not corresponding to the methodology. The investigation is specific about NSE, but no mention of this component is done in the title.
Review methodology of abstract. It appears to be lacking some phrases.
“divided into subgroups according to AHRE duration: Group 1: AHRE.”
Also mention in the abstract how was NSE assessed.
Please review the citations in the introduction, they are only referring to the last phrase. The authors should separate the references and not group all together in a single phrase.
The objective of the manuscript is not aligned with the title and methodology of the abstract.
Include a citation for the classification in the four groups. “Group 1: AHRE <5 minutes, Group 2: AHRE ≥5 minutes - <1 hour, Group 3: AHRE ≥1 hour - <24 hours, Group 4: AHRE ≥24 hours”
Were all the types of pacemaker included? Please, mention in the results the types of pace-makers that the patients had in the study.
Also, did any the patients have any problem with the pacemaker? How were these individuals assessed?
The authors should remove markers (•). It is a scientific manuscript. The exclusion criteria should be described in a single paragraph.
Full description of SPSS is needed.
How were the variables distributed? What were not normal? Include this information in statistical description.
Include Spearman correlation tests graphs as supplementary material.
How was the power of the study calculated? What was the study used as a reference for NSE? Please, include this information in the statistical section.
Please format the manuscript according to the instruction for authors. It is more easy for the reviewer to evaluate the manuscript. Tables and references are not according.
Why was the manuscript first uploaded in ResearchSquare?
https://www.researchsquare.com/
*This type of procedure blocks the iThenticate method of revision, and then it is impossible to assess plagiarism rates. If this happen again, it is recommended for the authors to continue their processing of the article at the same Publisher.
Reviewer 2 Report
Comments and Suggestions for Authors
I read with great interest this manuscript on pacemaker-detected atrial high rates and NSE levels. I have several points to improve the manuscript:
Major points:
1. I wonder about specificity of NSE levels as a marker of silent neurological ischemia. I suggest to downtone the conclusions that an association with silent ischemia was identified but rather that an association with a potential surrogate marker of ischemia was demonstrated.
2. LA dimension on echocardiography is an outdated parameter, could the authors provide LA volume indices and correlate them.
3. How many patients in both arms were on oral anticoagulation therapy?
4. The temporal association of AHRE and overt ischemia is obscure (see data from the ASSERT study). It would be interesting to measure NSE levels not only after AHRE episodes but also in periods without AHRE in the same patients.
Minor points:
1. Figure 1: I wonder what the second bar contributes, both bars (numbers and percentages) show the same information.
2. CHADS-VASC scores are used in patients with AF. AHRE are not AF unless demonstrated on an ECG. A risk score for a disease without the disease should thus be coined "virtual CHADS-Vasc score"
Round 2
Reviewer 1 Report
Comments and Suggestions for Authors
Satisfactory
Reviewer 2 Report
Comments and Suggestions for Authors
Unfortunately, the authors could not address all of my points but they gave explanations as to why the data are not available.